# SOURCE2SYNTH: SYNTHETIC DATA GENERATION AND CURATION GROUNDED IN REAL DATA SOURCES

## ABSTRACT

Large Language Models still struggle in challenging scenarios that leverage structured data, complex reasoning, or tool usage. In this paper, we propose *Source2Synth*: a new self-augmentation approach for teaching LLMs new skills that can be leveraged in low data regimes without relying on costly human annotations. . *Source2Synth* takes as input a custom data source and produces synthetic data points with intermediate reasoning steps grounded in real-world sources. *Source2Synth* improves the dataset quality by discarding low-quality generations based on their answerability. We demonstrate the generality of this approach by applying it to two challenging domains: we test reasoning abilities in multi-hop question answering (MHQA), and tool usage in tabular question answering (TQA). Our method improves performance by 25.51% for TQA on WikiSQL and 22.57% for MHQA on HotpotQA compared to the fine-tuned baselines.

## 1 INTRODUCTION

Large Language Models (LLMs) (Devlin et al., 2019; Chowdhery et al., 2022; Brown et al., 2020; Vaswani et al., 2017) have risen to popularity due to their remarkable ability to digest and generate human-like text (Radford et al., 2018). However, they still struggle with more complex tasks such as multi-step reasoning, tool use and manipulating or processing structured data. For many of these tasks there exists source data - for example, existing structured data on the web -, but little information of how to use these data to solve a task. In principle, one can achieve performance improvements during fine-tuning by enriching the data with human annotations collected for specific tasks. However, this is an expensive and time-consuming process (Touvron et al., 2023) subject to human-errors and bias.

In this paper, we propose *Source2Synth*, a general approach to produce *synthetic data grounded in external real-world sources*. Basing the data generation process on real-world sources steers the examples to be more realistic, diverse, and factually correct. We showcase our method on two challenging tasks: multi-hop questions based on sources from the web and tabular question answering using SQL as a tool. In both cases, models trained following *Source2Synth*'s pipeline achieve improved performance without relying on human annotations, resulting in a scalable data generation method for complex tasks, and present increased abilities at tackling corner cases.

*Source2Synth* consists of three stages: *Dataset Generation*, *Dataset Curation*, and *Model Finetuning*, see Figure 1. At the *Dataset Generation* stage, we start by selecting a data source (such as tables on the web or Wikipedia articles) to *ground* our synthetic generation in realistic information. Then, our method selects a seed topic to trigger the generation and condition all its components - for example a specific entity in a Wikipedia article or a factual statement about a table. Given the seed topic, the method then produces the full example: the instruction (e.g., question), the reasoning chain to arrive at the answer (e.g., the steps of multi-hop question answering, or tool use) and the answer itself.

At the *Data Curation* stage, the constructed synthetic dataset is split into two slices: the first half is used to fine-tune the LLM. We use this intermediate model to curate the second half of the synthetic dataset via an imputation and a filtering step by rejection sampling. For imputation, we blank some parts of the given example in order to get a more natural and cohesive entry. For filtering, we reject examples that cannot produce the correct answer in $k = 3$ tries. This provides a higher quality curated dataset for the second fine-tuning stage, resulting in a final better performing model on a given task.

To demonstrate the generality of our approach, we apply it to two different domains:

- Answering *tabular-based questions* by learning how to use SQL as a tool;
- Answering *multi-hop questions* by performing multi-step reasoning and information extraction.

To summarize, our key contributions are:

- We introduce a new method for generating synthetic examples aligned with the target task, given a real-world data source as context.
- We introduce a curation method based on filtering and imputation which yields higher quality data and improved task performance.

## 2  RELATED WORK

**Synthetic Data Generation using LLMs** A number of works propose different strategies to generate synthetic datasets leveraging pre-trained language models. Some of these works rely on knowledge-probing by first providing a prompt and letting the model either generate the continuation of a prefix or predict missing words in a close-style template (Schick & Schütze, 2020; Schick & Schütze, 2021; Petroni et al., 2019; Jiang et al., 2019). Other works introduce a variety of ways to improve the quality of synthetic data by using model-based or human filtering (Schick & Schütze, 2021; Liu et al., 2022; Li et al., 2024; Thoppilan et al., 2022). Our method however does not rely on human annotations, and we improve the quality of the synthetic data by leveraging the LLM itself. Furthermore, our selection of the seed topic is automated and we use real data as a starting point. We note that some recent work also leverages real-world data for specific cases, such as a corpus from the web to construct high-quality synthetic data (Nguyen et al., 2024) or open-source code snippets to generate diverse instruction data for code generation (Wei et al., 2024; Dubey et al., 2024). In our case, we proposes a general framework which can be applied across tasks and we do not require a back-translation approach or an initial finetuning to come up with the seed.
See Liu et al. (2024) for a thorough overview of synthetic data research and references therein.

**Teaching LLMs to Use Tools** Enabling LLMs to use different tools can extend their abilities towards manipulating structured data, retrieving information from external sources, or interacting with APIs. Even though the goal of our work is not specifically to teach models to use tools, but to develop a general synthetic data generation approach, we consider this to be a by-product. As an example, we demonstrate how our method can be applied so that LLMs use SQL. Various works augment LLMs with general tools or API calls (Parisi et al., 2022; Schick et al., 2023; Tang et al., 2023), while some propose to interweave intermediate reasoning steps with API calls (Gao et al., 2023; Cai et al., 2024; Paranjape et al., 2023) which improves performance on more complex tasks. Finally, handling unseen tools at test time has also been tackled (Paranjape et al., 2023; Mekala et al., 2024). See Mialon et al. (2023) and Qin et al. (2023) for an in-depth review of augmented tool-use.

**Teaching LLMs to use SQL** The above approaches usually tool usage is restricted to inputs that are strings or numbers. However, using structured data (like tables and graphs) during post-training can be useful to enhance the LLM's capabilities in complex tasks. A particular tool of interest is SQL since it enables aggregating information from tabular data. There exist a variety of benchmarks that have been proposed to assess LLMs abilities to generate SQL as well as their performance on tabular-based question answering leveraging SQL tasks (Li et al., 2023a; Zhong et al., 2017). Alternatively, handling tabular data directly by LLMs has also been tried (Herzig et al., 2020; Gemmell & Dalton, 2023), and tabular question answering benchmarks have been proposed (Pasupat & Liang, 2015).

## 3  METHOD

*Source2Synth* produces high-quality synthetic examples grounded in external real-world data sources, which can be fed to the LLM as step-by-step examples at fine-tuning. *Source2Synth* is composed of three stages: *Dataset Generation*, *Dataset Curation*, and *Model fine-tuning*.

### 3.1  DATASET GENERATION

**Data source selection** The generation process begins by selecting a data source. This can be an already existing dataset re-purposed for a given task, a collection of existing data points that we would

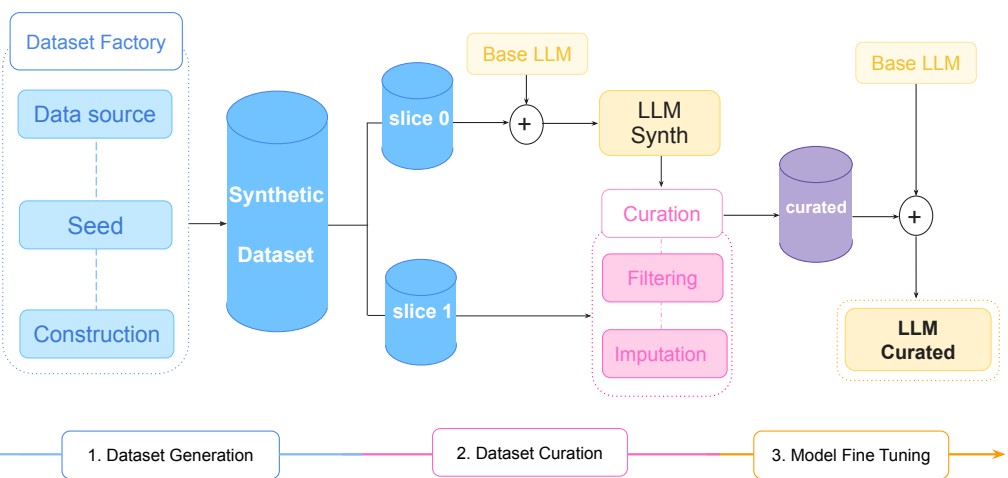

Figure 1: **Overall *Source2Synth* Method.** In the *Dataset Generation* step we first choose a data source to build our dataset from. For each example we select a seed topic to condition the generation on, and use the data source and seed together to construct the example. The resulting synthetic dataset is sliced in two: slice 0 is used to fine-tune an intermediate version of the LLM (*LLMSynth*), and we use *LLMSynth* to curate slice 1 through filtering and/or imputation during the *Dataset Curation* step. The resulting curated dataset is of higher quality and aligned with the user's design. At the *Model Finetuning* stage, the final LLM (*LLMCurated*) is trained on the curated synthetic dataset, which can then be used to provide good performance on the task of interest.

like to leverage to construct a new dataset, or structured information (e.g. graphs, tables). There is no need for human annotations on the entries, as *Source2Synth* will enrich it with extra instructions.

**Seed** In order to create a given example of our new synthetic dataset, we first generate a *seed* topic as the initial trigger for the generation process, which is chosen conditioned on a randomly selected portion of the source data. The seed inspires the creation of the entry and dictates how the source data will be used. In addition, the randomness of the seed ensures variety in the generated data.

**Dataset construction** In order to tackle complex tasks, LLMs can leverage a step-by-step approach (Wei et al., 2022) that divides reasoning into smaller sub-tasks plus instructions on how to merge back each step into the final one. In *Source2Synth*, we leverage the seed to build synthetic data step-by-step, decomposing into such intermediate steps in order to arrive at an answer for a given question. This reasoning chain can then be used as supervision by providing it as the target in the synthetically generated training examples.

## 3.2 Dataset Curation

During curation, *LLMSynth* is then used to improve the quality of the second slice of the dataset using imputation plus a filtering step. We observe that tasks that require compositionality during the construction of the synthetic entry benefit from imputation since it allows the LLM to reformulate into something of higher likelihood under the model's distribution. For example, in MHQA, the merging of two sub questions might produce a multi-hop question that sounds artificial. Following the effort in aligning LLMs with human preferences for a more natural use, we observe by looking at the perplexity of imputation that partially reconstructing $Q$ leads to more human-like questions.. After these steps, we obtain the final curated dataset (shown in purple in Figure 1).

**Data filtering** During filtering, *LLMSynth* is used to predict the output of the given synthetic example using $k = 3$ tries. If the output cannot be predicted at least once, it is assumed the example is low quality and is not included in the final curated dataset.

**Data Imputation** We also consider an imputation process, which involves blanking parts of the augmented data points and using the LLM to fill in the blanks, to replace those fields. This is to provide cleaner data which is less unnatural.

### 3.3 MODEL FINE-TUNING

At this stage, we fine-tune on the curated synthetic dataset, initializing from a base or instruction-tuned version of the LLM. We use our dataset for supervised training of both the reasoning chain and the final answer. The resulting model *LLMCurated* is then ready to perform the desired task.

## 4 SOURCE2SYNTH'S APPLICATIONS

The general pipeline described above can be used to produce examples for the task at hand and to teach LLMs new skills. To demonstrate the impact of *Source2Synth*, we apply it to two challenging tasks where LLMs struggle which are also areas of great interest for the community: multi-hop question answering and tabular question answering.

### 4.1 MULTI-HOP QUESTION ANSWERING

In multi-hop question answering (MHQA), we generate a dataset of multi-hop question-answer pairs, enriched with the reasoning chain that is used to answer the question. The chain consists of question decomposition into sub questions with answers, plus the entity that links them.
See Figure 2 for an overview of the procedure and Figure 3 - Right for an example response from the model that underwent the *Source2Synth*'s pipeline.

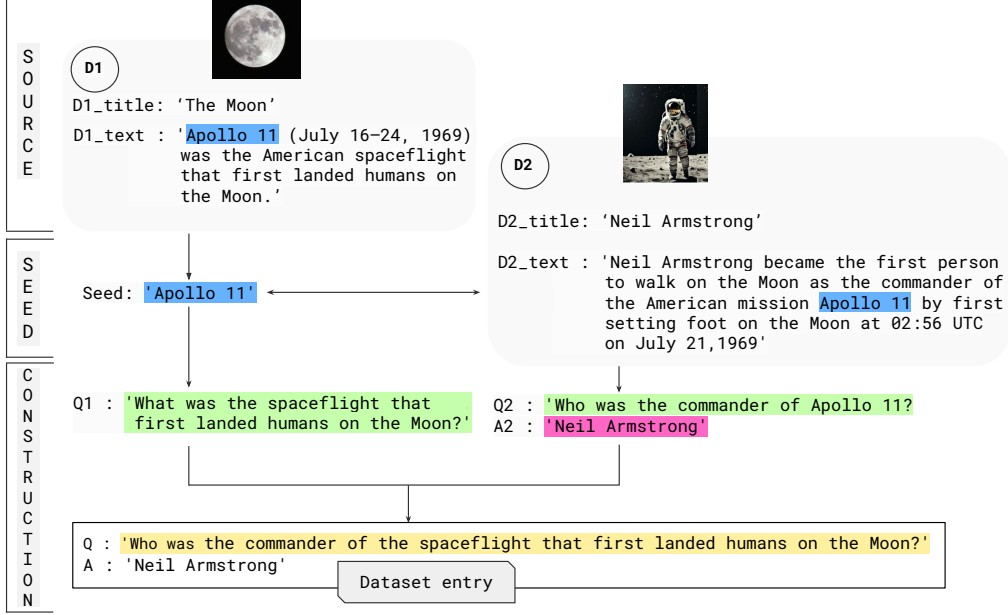

Figure 2: *Source2Synth* **synthetic data generation process for multi-hop question answering**. The method first randomly picks one article $D_1$, in this case with title "The Moon". At the *Seed* stage, an entity E is selected from $D_1$'s pool of entities, "Apollo 11". Then, documents are sampled from the related documents pool of $D_1$ such that $E$ is present, and $D_2$, "Neil Armstrong", is selected. A question $Q_1$ is then generated from $D_1$ with the constraint that the answer $A_1$ is the entity itself. A second question $Q_2$ is then generated from $D_2$ with the constraint that its main topic is the entity. We then prompt an LLM to merge the two questions based on the link/entity they have in common to produce the final question, reasoning chain and answer that comprise the training example.

### 4.1.1 DATASET GENERATION

**Data source selection** For multi-hop question answering, we pick English Wikipedia (Wikipedia contributors, 2004) as the data source, since it contains articles in natural language as well as additional meta-information like links to related articles. The data generation process starts by randomly selecting an initial article, denoted as $D_1$, among all available Wikipedia articles. For each $D_1$ we collect $n \geq 2$ related articles.

**Seed** An MHQA seed topic corresponds to an entity $E$ retrieved from $D_1$. The seed in MHQA doubles also as the "hop" in the multi-hop question $Q$ that we aim to generate, since $E$ links the $n = 2$ subquestions that compose $Q$. For example, in Figure 2, we sample "The Moon" article at random, denoted by $D_1$, and the corresponding entity, denoted by $E$, is "Apollo 11" (displayed in blue). Then, we pick "Neil Armstrong" as $D_2$ from the pool of related articles, since it contains a paragraph where the entity "Apollo 11" is included.

**Dataset construction** We prompt an instruction-tuned language model to generate two questions: a question $Q_1$ based on $D_1$ and whose answer is the selected entity $E$, and a second question $Q_2$ based on $D_2$ such that its main topic is $E$. See Figures 18 and 19 for the exact prompts. For example, in Figure 2, $Q_1 = $ "What was the spaceflight that first landed humans on the Moon?", the hop is $E = $ "Apollo 11" and $Q_2 = $ "Who was the commander of Apollo 11?". We then prompt the LLM to merge the two questions, in order to generate the final two-hop question $Q$ by using the entity as a conceptual link (hop). The exact prompt is given in Figure 17. For MHQA, we generated a total of 1250 synthetic questions starting from a collection of 50 Wikipedia articles (randomly selected).

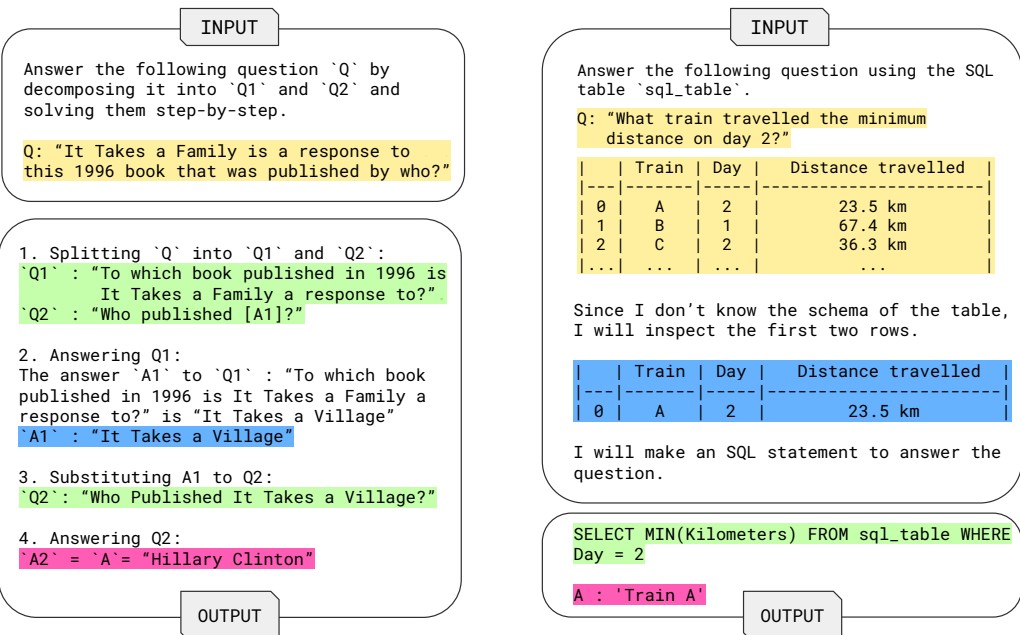

Figure 3: **Left: Example *Source2Synth* Response on MHQA** (closed book inference). We show the model's response (reasoning steps and answer) to a multi-hop input question (yellow). The colours highlight the generation of the corresponding augmented entries: the decomposition into sub questions $Q_1$ and $Q_2$ in green, the seed $A_1$ in blue, and the final answer $A_2$ in red.
**Right: Example *Source2Synth* Response on TQA.** We show the model's response (SQL call and final answer) to the tabular input question (yellow). The coloured parts highlight the generation of the corresponding augmented entries: SQL in green, and the final answer in red.

### 4.1.2 DATASET CURATION

In the MHQA experiments, the curation step removed around 13% of the questions originally generated.

**Data filtering** We check if the predicted answer matches the answer in the synthetically generated example, and if after $k = 3$ tries tries the LLM has not supplied the correct answer we filter out the entry entirely. See Figure 3 - Left for an example of model inference.

**Data Imputation** For MHQA, we blank $Q_1$ and provide the LLM with $Q$, $Q_2$, $E$, and the relative doc sample $D_1$ as context when asking it to reconstruct $Q_1$. The new candidate $Q_1'$ for $Q_1$ is then assessed: if $A'$ (the answer to the new multi-hop question $Q'$ resulting from piecing together $Q_1'$ and $Q_2$) matches $A$ (the original answer to $Q$) then we keep the example. We find that asking the model to reconstruct parts of the multi-hop question in-context results in a more natural and cohesive question, thus removing some of the unnaturalness of the text that can occur from automatically generated and merged examples (see Appendix for more details).

## 4.2 TABULAR QUESTION ANSWERING

In Tabular question answering (TQA) we generate a question-answer dataset where each question is based on a table from the data source. Generated training examples are hence enriched with annotations built from automatically-generated interesting facts retrieved from the table.

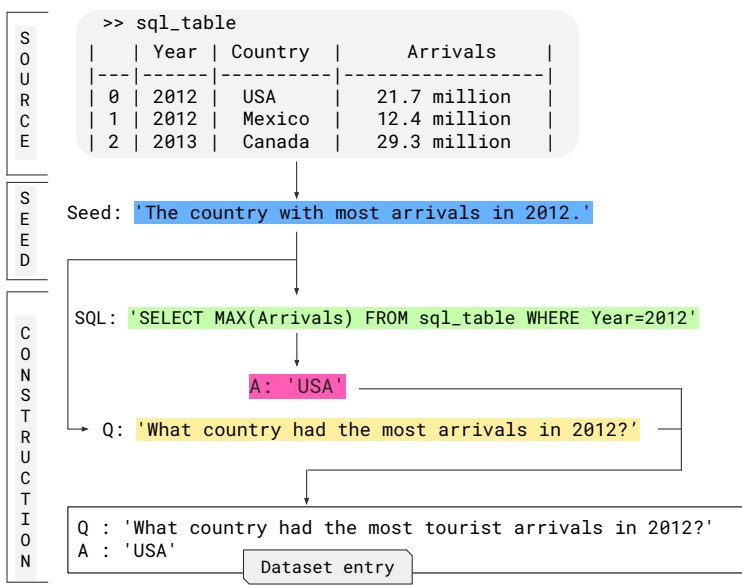

Figure 4: **Source2Synth synthetic data generation process for Tabular question answering.** The method first generates the seed, which is a fact based on the table (shown in blue). Given the seed and table, an *SQL query* is then generated (in green) as well as its translation into natural language (the question $Q$). Then the SQL is executed on the table to obtain the answer **A**.

### 4.2.1 DATASET GENERATION

**Data source selection** In the TQA case, we use 4k unlabeled tables from the WikiSQL training dataset as sources (Zhong et al., 2017).

**Seed** We then prompt an instruction-tuned language model to generate a statement based on the table. This statement corresponds to our seed topic for the generation and is a pertinent interesting fact or set of observations in natural language that can be derived from the table. The prompt is given in Figure 13.

**Dataset construction** We next generate an SQL-statement by zero-shot prompting the LLM: we provide the table and the seed (factual statement) as context, see Figure 14 for the exact prompt. Given the produced SQL statement, it is then executed using the Python library *sqlite3*[1] to obtain an SQL answer formatted as a table. If the generated statement is invalid, we discard it and re-generate.

---

[1]https://www.sqlite.org

We generate a total of 10k SQL statements based on the source tables. Checking the statements for validity (i.e. refusing non-executable SQL statements) brings the dataset size to 8k (per slice).

### 4.2.2 DATASET CURATION

In Tabular QA, the curation process consists only of the filtering step. After curation, we keep 2160 (27%) of the original examples in slice 1.

**Data filtering** We check if the predicted answer of *LLMSynth* fine-tuned on slice 0 matches the answer in the synthetically generated example, and if after $k = 3$ tries the model has not supplied the correct answer we filter out the entry entirely. See Figure 3 - Right for an example of model inference.

## 5 EXPERIMENTAL SETUP

We test our method on two domains: *tabular question answering* and *multi-hop question answering*. For each, we use *Source2Synth* to generate and curate a high quality dataset suitable for fine-tuning, and compare our method to a number of baselines.

### 5.1 MULTI-HOP QA SETUP

**Data** To evaluate *Source2Synth* on MHQA, we evaluate it on HotpotQA (Yang et al., 2018): a benchmark based on Wikipedia containing 113,000 examples of multi-hop QA pairs, split in train, test, and validation sets.

A comparison question entails comparing the same concept between $n$ objects (e.g. "Who is the tallest student in class?"), while a bridge question builds on a logical and/or causal link and requires deriving statements to get to the answer (e.g. "What is the height of the student that topped the entry exam?" - this requires first identifying the student that topped the exam). The hop length is the number of comparison objects for comparison questions or the number of links for bridge questions. In our case, we chose $n = 2$ to be consistent with HotpotQA . The test set consists of 7,405 entries, split evenly between bridge and comparison questions. We only generate synthetic data for bridge questions, since they pose a bigger challenge to current LLMs and to counterbalance this disparity, we include 500 comparison questions from HotpotQA 's training dataset in our fine-tuning dataset.

**Metrics** We measure the performance using soft exact match (soft-EM) as the metric. Soft-EM is 1 if the generated output contains the golden answer and 0 otherwise.

**Model** In MHQA experiments we chose Llama-2 70B-Chat and we fine-tune *Source2Synth* and various other baseline methods initializing from this model. *Source2Synth* is trained with 1250 synthetic examples, unless noted otherwise, in addition to the 500 HotpotQA examples above.

**Baselines** We compare *Source2Synth* to the following baselines:

- Instruction-tuned LLM: using LLama 2 70B-Chat for the task in a zero-shot manner.
- Fine-tuned LLM (HotpotQA only): fine-tuning from the base model on 500 HPQA examples from the training split.
- *LLMSynth* (Synthetic dataset only): training our model with 1250 synthetic examples from Slice 0 (see Figure 1), *without* the data curation step.
- *LLMSynth* (Synthetic and HotpotQA ): training with the uncurated synthetic data in addition to the 500 HPQA examples.

For all the models listed, we tested them using two prompting methods: a zero-shot and a three-shot CoT prompt, see the Appendix E for details.

### 5.2 TABULAR QA SETUP

**Data** We conduct evaluations with the WikiSQL (Zhong et al., 2017)'s validation split. WikiSQL consists of a corpus of 80,654 hand-annotated examples of natural language questions, SQL queries,

and SQL tables created from 24,241 tables extracted from Wikipedia. The validation split contains 7,857 examples after removing non-executable SQL tables, see Appendix B for more details.

**Metrics** We measure performance using the exact match (EM) and the soft-EM metrics. The EM metric equals 1 if the golden answer is equal to the generated answer and 0 otherwise.

**Model** For TQA, we use the Starchat-beta language model Li et al. (2023b) from Huggingface as the initial language model (batch size 32, 100 steps, lr 0.0001, linear warm-up). The Starchat model is an instruction-tuned LLM with 16 billion parameters trained to act as a helpful coding assistant. This model is a fine-tuned version of StarCoder Li et al. (2023b), a LLM which was pre-trained and then fine-tuned on a large code corpus, which contains SQL statements, and successively fine-tuned on 35B Python tokens.

**Baselines** We compare the performance of our *Source2Synth* method against a variety of baselines. The baselines consist of prompting the Starchat-beta instruction-tuned language model as follows:

- *Zero-shot Table QA*: prompt with the task instruction, the table and the question in a zero-shot fashion. See Figure 9 for the prompt.
- *One-Shot No Context QA*: prompt with the task instruction and a one-shot example containing a question and answer, together with the actual question for the model to answer. See Figure 10 for the prompt.
- *One-Shot Table QA*: prompt that includes the table for both the one-shot example and the question to be answered. We use one-shot due to LLM context length and the typically large size of the tables. See Figure 11 for the prompt.
- *One-shot Table+SQL QA*: the prompt includes an example containing the table and question, and an instruction suggesting that the model can leverage an SQL tool. We then execute the predicted SQL to obtain the answer. See Figure 12 for the prompt.
- *LLMSynth*: Fine-tune the model with synthetic data *without* applying the data curation step.

## 6 RESULTS

### 6.1 MULTI-HOP QUESTION ANSWERING

Table 1: **Evaluation of *Source2Synth* on Multi-hop question answering.** The models shown are fine-tuned with 500 entries from HotpotQA ('HotpotQA ") and/or 1250 entries from the *Source2Synth* Synthetic Dataset ("Synthetic Dataset"). Using *Source2Synth* curated synthetic data in combination with HotpotQA (last row) works best.

| Method | 0-shot | 3-shot CoT prompt |
|---|---|---|
| Instruction-tuned LLM (LLama 2 70B-Chat) | 40.45% | 44.13% |
| fine-tuned LLM (HotpotQA only) | 53.22% | 58.40% |
| *LLMSynth* (Synthetic dataset only) | 52.31% | 56.70% |
| *LLMSynth* (Synthetic and HotpotQA ) | 57.46% | 62.73% |
| *LLMCurated* (Synthetic and HotpotQA ) | 65.23% | 66.05% |

Table 2: **Analysis of MHQA bridge and comparison questions with respect to level of difficulty.** We evaluate models on the full train dataset (where questions are labelled with easy, medium and hard). *Source2Synth* outperforms both the baseline and the model fine-tuned on HotpotQA , yielding an LLM capable of handling hard questions of both types.

| Model | Bridge | | | Comparison | | |
|---|---|---|---|---|---|---|
| | **Hard** | **Medium** | **Easy** | **Hard** | **Medium** | **Easy** |
| Llama2-70B-Chat | 14.5% | 27.2% | 30.1% | 66.6% | 71.3% | 73.2% |
| Fine-tuned LLM (HotpotQA only) | 20.1% | 29.8% | 34.3% | 74.5% | 78.3% | 82.1% |
| LLMCurated-1250 | 31.3% | 35.6% | 39.7% | 83.1% | 85.7% | 87.8% |

**Overall performance of *Source2Synth* on MHQA** We report the experimental results in Table 1. We include the baselines of the vanilla instruction-tuned LLM (0-shot and 3-shot, please see Prompt 16), a *fine-tuned LLM* using only the HPQA 500 examples from the train split (second row), and *LLMSynth* which only uses the uncurated synthetic data for fine-tuning (third row). All fine-tuned methods outperform the instruction-tuned model (first row). Using only synthetic data or only HotpotQA data for fine-tuning demonstrates worse performance than when combined, whether the synthetic data is curated, as in *LLMCurated* (fifth row) or not, as in *LLMSynth* (fourth row). Once we use the full *Source2Synth* pipeline to obtain the curated synthetic dataset for fine-tuning we see further performance improvements *LLMCurated* (fifth row) over not curating the data (fourth row).

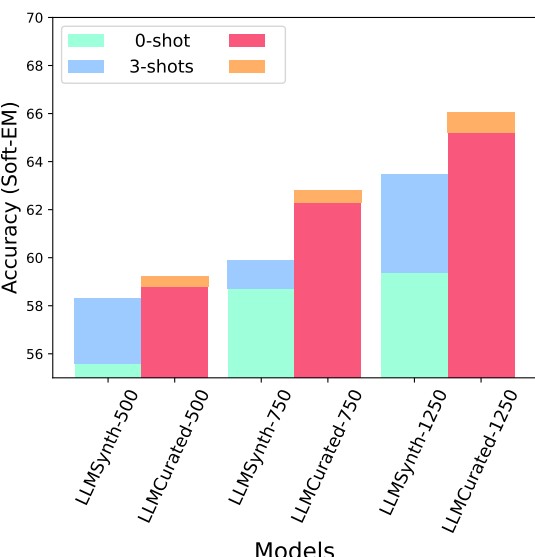

Figure 5: **Synthetic Data scaling performance.** We show how the performance of *Source2Synth* changes with respect to MHQA data mix size, both before and after curation. During the curation step, the following percentages of samples were removed: 7% for 500, 8% for 750, 11% for 1250. *LLMSynth* (before curation) performs worse than *LLMCurated* (after curation) despite having more samples – but both approaches improve with more data.

**Analysis of performance on different question types and levels of difficulty** We study the capabilities of our model by analysing the performance of LLM-Curated-1250 with particular focus on the type and difficulty of the questions – namely hard/medium/easy bridge and comparison questions. We compare the performance of the base model, the model fine-tuned on HotpotQA, and *Source2Synth* according to the difficulty level, as provided by the HotpotQA train dataset. We also subdivide the results according to the type of question (bridge vs. comparison). Results are given in Table 2.

We observe that *Source2Synth* performs better across all types of questions and difficulties, with an average overall gain of 12.4% on the base LLM and a 7.5% gain compared to the LLM fine-tuned on HotpotQA. In particular, by applying our method, the resulting model is able to achieve +16.8% and +16.5% on hard bridge and comparison questions respectively, when comparing to the baseline. Furthermore, it is interesting to see substantial improvement on comparison-type questions, despite not explicitly targeting those during synthetic generation. Hard questions pose a greater challenge to the reasoning abilities of LLMs and these results introduce *Source2Synth* as a possible method for further improvement.

**Scaling performance** Since *Source2Synth* can be leveraged when the amount of available data is low, we also report scaling performance in Figure 5. We study how performance evolves when adding more synthetic data in the fine-tuning data mix - that already includes 500 samples from the HPQA train split. We perform the analysis on *LLMSynth* and *LLMCurated* to show the impact of the curation technique. In both cases and in all data mixes, we see that applying the *Source2Synth* pipeline results in a stronger model on the task. For the *LLMSynth* model fine-tuned on uncurated samples we see that providing more synthetic examples leads to a steady improvement in performance across all data

Table 3: **Tabular question answering.** The models are fine-tuned using *Source2Synth* curated synthetic data only. Performance comparison on the WikiSQL evaluation dataset.

| Method | Exact Match | Soft-EM |
|---|---|---|
| One-Shot No Context QA (Starchat-beta LLM) | 0.25% | 16.22% |
| Zero-shot Table QA (Starchat-beta LLM) | 1.83% | 20.07% |
| One-Shot Table QA (Starchat-beta LLM) | 2.03% | 31.06% |
| One-shot Table+SQL QA (Starchat-beta LLM) | 12.30% | 34.13% |
| *LLMSynth* (Synthetic dataset only) | 23.86% | 34.21% |
| *LLMCurated* (Synthetic dataset only) | 34.50% | 42.80% |

sizes, for both zero-shot and three-shot prompting variants. *LLMCurated* follows a similar trend, but consistently outperforms the uncurated version of the model, for all training set sizes. Overall, we observe that using our synthetic data generation pipeline to construct more data brings further performance gains in the task.

## 6.2 TABULAR QUESTION ANSWERING

We report the experimental results for Tabular question answering in Table 3. Firstly, we see that providing no context about the table when prompting the instruction-tuned StarChat language model has very poor performance (first row), with an EM metric of 0.25%. This is expected, since questions in WikiSQL require information contained in the table to answer, while the model does not have any other information except for the general knowledge stored in its parameters. However, even if we pass the table as part of the prompt, the performance does not improve much due to its difficulties to digest structured data. For example, passing in a zero-shot fashion (second row) only has an EM metric of 1.83%. While passing an example table usage in a one-shot fashion (third row) improves the soft-EM metric, the EM metric is still very low (2.03%). Hence, this is still very challenging for the model. Thirdly, the performance increases once we provide a one-shot example containing the relevant table and SQL query (fourth row), with an EM of 12.3%. The ability to use the SQL tool improves performance markedly.

We obtain a significant increase in performance when we fine-tune the StarChat model using the *Source2Synth* curated data (last row), with an EM of 34.5%. Our full method performs significantly better than fine-tuning the StarChat language model using synthetic data without curation, *LLMSynth* (second to last row) which has an EM of 23.86%, although that still outperforms the other baselines by a large margin as well, indicating the utility of our *Source2Synth* synthetic data generation scheme.

## 7 LIMITATIONS

In this paper, our applications use a single seed or table per query to derive questions. However, *Source2Synth* can be extended to more complex scenarios e.g. multiple hops or queries that require multi-table tool-use. This can be done by looping the dataset generation steps and feeding the result of the previous step as input to the next one. Our method could also be improved with more clever sampling techniques. We consider this to be an interesting avenue of future research.

## 8 CONCLUSION

In this paper, we introduce *Source2Synth*, a new method for generating and curating high-quality synthetic data grounded in real data sources. We demonstrate its utility on two tasks that pose significant challenges for LLMs: multi-hop reasoning and tabular question answering with SQL. Our work could also be beneficial in other low-data regimes and on other tasks and in diverse fields.

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

# A  MORE RESULTS ON PROMPT ENGINEERING

| Prompt Type | Model Accuracy (soft-EM) |
|---|---|
| 0-shot | 40.45% |
| Role | 22.34% |
| 1-shot | 26.65% |
| Few-shots (5-shots) | 21.83% |
| Role (1-shot) | 28.29% |

Table 4: **MHQA prompts sweep.** Overview of the model's accuracy across different prompt strategies. *Role* "You are a QA-robot. Answer the following question:". Model used: Llama-2-70B-Chat, Dataset: HotpotQA test.

# B  SQL NON-EXECUTABLE CODE FILTERING

We discard incorrect SQL statements - i.e. whose execution with *sqlite3*[2] leads to an error. Discarded proportion: out of 50 tables, we generate 800 seed statements and the number of valid (executable) SQL statements was 658.

# C  ILLUSTRATED EXAMPLE: ADAPTING TO A NEW TASK

There are (at max.) three components in *Source2Synth*'s pipeline that need to be changed in order to adapt it for a new task: prompts, seed, and data source. If the new task builds on document- or table-use, some parts of the two applications showcased in the main body of the paper can be reused. We proceed illustrating how to adapt prompts, seed, and data source in case of the following new task: generating code (Python functions) to compute statistics on spreadsheets. Since this task builds on tables, we integrate TQA in it by leveraging its seed generation system. We believe that this is a meaningful and diverse example as it showcases a shorter tasks that does not produce a QA dataset and that leverages structured data. Steps for adaptations:

1. *Select the data source* The data source is the collection of spreadsheets of interest to the user. This can be a custom dataset, or a public one (like WikiSQL).;

2. *Select the seed* It is important to condition the seed generation so that it reflects the goal of the user. For example since we would like to compute statistics on the tables in the dataset, it is important to define which statistics are of interest to the user (max / min value, average, median, etc...);

3. *Changing the seed generation prompt* so that it conditions the generation of the statement to focus on the interests of the user. For example, in practice we would update the prompt in Fig.13 as follows - please see Fig. 6;

4. *Using the seed to generate code* Adapt the data generation prompt to output Python functions biased on the statement produced - please see Fig. 7.

---

[2]https://www.sqlite.org

---

**Generating a seed in the new task.**

Please generate an interesting statement about this table. The statement is a fact about one of the columns in the following table. The statement should include one of these metrics: {statistics_of_interest_list}

{table}

As a result of this, an interesting statement about the table and the metrics is:

---

Figure 6: Prompt adaptation for seed generation (new task)

---

**Generating code in the new task.**

Please generate a Python function based on the table and statement below. The function must solve the task described by the statement.The table is an input variable to the function. {table}

Statement: {seed}

As a result of this, a Python function that solves the problem described by the statement is:

---

Figure 7: Prompt adaptation for output generation (new task)

## D    ILLUSTRATED EXAMPLE: ADAPTING TO A NEW DOMAIN

Similarly to across tasks, there are three components to adapt for a new domain application: prompts, seed, and data source. We illustrate an example of QA task in a medical domain. Steps for adaptations:

1. *Select the data source* The data source is a collection of medical documents (either private or public like PubMed [3]);

2. *Generating the QA dataset* Since the task is the same as MHQA (but just applied to a different domain - medical), once the data source is adapted we can leverage the same pipeline to generate the seed and QA pairs. Please see Fig. 2 the seed and construction stages for a visual reminder of the pipeline, and 18, 19 for the prompts;

---

**Comparing Q pre- and post- imputation**

*Before imputation:*
$Q$: "What pet did the poet and father of mathematician Ada Lovelace had when he was a student at Trinity out of resentment for rules forbidding pet dogs like his beloved Boatswain?"
$Q_1$: "What pet did the poet Lord Byron had when he was a student at Trinity out of resentment for rules forbidding pet dogs like his beloved Boatswain?"
$Q_2$: "Who is the father of mathematician Ada Lovelace?"
$E$: "Lord Byron"
$A$: "A bear"
$D_1$: "Lord Byron also kept a tame bear while he was a student at Trinity out of resentment for rules forbidding pet dogs like his beloved Boatswain."

*After imputation:*
$Q'$: "What pet did the poet and father of mathematician Ada Lovelace had when he was a student at Trinity?"
$Q'_1$ : "What pet did the poet Lord Byron had when he was a student at Trinity?"

---

Figure 8: Comparing Q pre- and post- imputation

## E    PROMPTS USED IN OUR EXPERIMENTS

---

[3]https://huggingface.co/datasets/ncbi/pubmed, https://pubmed.ncbi.nlm.nih.gov/about/

**Zero-shot Table QA prompt.**

Answer the following question using the table below. You may leverage an SQL tool.

{table}

Q: {question}

Figure 9: Zero-shot Table QA prompt for the TQA task.

**One-Shot No context QA prompt.**

– Example –
Q: What was the last year where this team was part of the US A-league?
A: 2004

Now do the same for the following question.
Q: {question}

Figure 10: One-Shot No context QA prompt for the TQA task.

**One-shot Table QA prompt.**

```
-- Example --
Answer the following question using the table below.
Your answer should be short and concise.

Season | Team         | League_apps | Goals
1923   |Swindon Town  | 55          | 3
1922   |Swindon Town  | 14          | 4
1921   |Swindon Town  | 24          | 11
1920   |Swindon Town  | 26          | 16
1919   |Swindon Town  | 20          | 10
1914   |Swindon Town  | 23          | 12
1913   |Swindon Town  | 24          | 18
1912   |Swindon Town  | 12          | 9
1911   |Swindon Town  | 20          | 16
1910   |Swindon Town  | 30          | 19
1909   |Swindon Town  | 33          | 19
1908   |Swindon Town  | 34          | 28
1907   |Swindon Town  | 30          | 17
```

Q: How many league appearances were there between 1907 and 1909 (inclusive)?
A: 97

Now do the same for the following table and question.

{table}

Q: {question}

Figure 11: One-shot Table QA prompt for the TQA task.

---

**One-shot Table+SQL QA prompt.**

```
-- Example --
Answer the following question using the table below.
You may leverage an SQL tool.
The table is stored in a variable 'sql_table' and has the following schema:

Season | Team         | League_apps | Goals
1923   |Swindon Town | 55          | 3
1922   |Swindon Town | 14          | 4

Q: How many league appearances were there between 1907 and 1909 (inclusive)?

SQL: SELECT SUM(League_apps) FROM sql_table WHERE Season BETWEEN 1907 AND 1909

       | Result
result | 97

Now do the same for the following table and question.
```

{table}

Q: {question}

---

Figure 12: One-shot Table+SQL QA prompt for the TQA task.

---

**Generating a seed in TQA.**

Please generate an interesting statement about this table. The statement is a fact about one of the columns in the following table.
{table}

An interesting statement as a result of this is:

---

Figure 13: Prompt used to induce a pertinent and interesting seed topic in TQA. This is done zero-shot.

---

**Generating meaningful SQL in TQA.**

Please generate SQL statements for the following table:

{table}

Seed: {seed}

An interesting SQL statement as a result of this is

---

Figure 14: Prompt used to induce a meaningful SQL statement given the table and seed for the TQA task. This is done zero-shot.

---

**Generating a question in TQA.**

I want to convert an SQL statement into a question.
Here is the original table:
{table}

SQL: {SQL}

What is the question that this SQL statement would be the answer to?

---

Figure 15: Prompt used to induce a meaningful question using the table and generated SQL query for the TQA task. This is done zero-shot.

---

**Three-shot CoT prompt used at evaluation time on MHQA.**

Answer the following multi-hop question 'Q' by decomposing it into 'Q1' and 'Q2' and solving them step-by-step. Learn from the following 3 examples. As shown in the following example:

-- Example #1 --
'Q' = 'Who was the commander of the spaceflight that first landed humans on the Moon?'

1. Splitting 'Q' into 'Q1' and 'Q2':
'Q1' : 'What was the spaceflight that first landed humans on the Moon?';
'Q2' : 'Who was the commander of [A1]?';

2. Answering Q1:
The answer 'A1' to 'Q1' : 'What was the spaceflight that first landed humans on the Moon?' is 'Apollo 11'. 'A1' = 'Apollo 11'

3. Substituting A1 to Q2:
'Q2' : 'Who was the commander of Apollo 11?',

4. Answers Q2:
The answer 'A2' to Q2 : 'Who was the commander of Apollo 11?' is 'Neil Armstrong'.
'A2' = 'A' = 'Neil Armstrong'

-- Example #2 --
'Q' = 'What is the main ingredient in the flagship product of Ferrero?'

1. Splitting 'Q' into 'Q1' and 'Q2':
'Q1': 'What is the flagship product of Ferrero?'
'Q2': 'What is the main ingredient in [A1]?'

2. Answering Q1:
The answer 'A1' to 'Q1' : 'What is the flagship product of Ferrero?' is Nutella'.'A1' = Nutella'

3. Substituting A1 to Q2:
'Q2' : 'What is the main ingredient in Nutella?',

4. Answers Q2:
The answer 'A2' to Q2 : 'What is the main ingredient in Nutella?'.
'A2' = 'A' = 'Hazelnuts

--Example #3 --

'Q' = 'Who was the Roman Emperor when Jesus was born?'
1. Splitting 'Q' into 'Q1' and 'Q2':
'Q1': 'When was Jesus born? '
'Q2': 'Who was the Roman Emperor in [A1]?'

2. Answering Q1:
The answer 'A1' to 'Q1' : 'When was Jesus born?' is 1 BCE. 'A1' = 1 BCE

3. Substituting A1 to Q2:
'Q2' : 'Who was the Roman Emperor in 1 BCE?',

4. Answers Q2:
The answer 'A2' to Q2 : 'Who was the Roman Emperor in 1 BCE?'.
'A2' = 'A' = 'Caesar Augustus'

You MUST apply this structure when asked to answer a multi-hop question 'Q'. Now answer the multi-hop question 'Q' as shown in the examples above.
Q: {question}

Figure 16: *Three-shot CoT prompt* used at evaluation time in MHQA.

---

**Prompt used to merge Q1 and Q2 in MHQA.**

Merge 'Q1' and 'Q2' into a single multi-hop bridge question 'Q'.
Learn from the following 3 examples. As shown in the following example:

-- Example #1 --

'Q1' : "What was the spaceflight that first landed humans on the Moon?"
'Q2': "Who was the commander of Apollo 11?"

Solution:
1. Answer Q1; 'A1' is "Apollo 11"
2. If 'A1' is in 'Q2' print(A1); 'A1' = Apollo 11 is in 'Q2' so I print "Apollo 11"
3. Since you found 'A1' in 'Q2', rewrite 'Q2' so that you delete 'A1' and substitute 'Q1' there;
Rewriting Q2. Original 'Q2': "Who was the commander of Apollo 11?". Since 'A1' is in 'Q2', I delete it and write 'Q1' there. Rewritten 'Q2': "Who was the commander of the spaceflight that first landed humans on the Moon?"

The single multi-hop question is therefore the rewritten 'Q2'.
'Q2' = 'Q' = "Who was the commander of the spaceflight that first landed humans on the Moon?"

-- Example #2 --

'Q1': What is the flagship product of Ferrero?
'Q2': What is the main ingredient in Nutella?
Solution:
1. Answer Q1; 'A1' is "Nutella"
2. If 'A1' is in 'Q2' print(A1); 'A1' = "Nutella" is in 'Q2' so I print "Nutella"
3. Since you found 'A1' in 'Q2', rewrite 'Q2' so that you delete 'A1' and substitute 'Q1' there;
Rewriting Q2. Original 'Q2': "What is the main ingredient in Nutella?".
Since 'A1' is in 'Q2', I delete it and write 'Q1' there.
Rewritten 'Q2': "What is the main ingredient in the flagship product of Ferrero?"

The single multi-hop question is therefore the rewritten 'Q2'. 'Q2' = 'Q' = "What is the main ingredient in the flagship product of Ferrero?"

-- Example #3 --

'Q1': "When was Jesus born?"
'Q2': "Who was the Roman Emperor in 1 BCE?"

Solution:
1. Answer Q1; 'A1' is "1 BCE"
2. If 'A1' is in 'Q2' print(A1); 'A1' = 1 BCE is in 'Q2' so I print "1 BCE"
3. Since you found 'A1' in 'Q2', rewrite 'Q2' so that you delete 'A1' and substitute 'Q1' there;
Rewriting Q2. Original 'Q2': "Who was the Roman Emperor in 1 BCE?". Since 'A1' is in 'Q2', I delete it and write 'Q1' there. Rewritten 'Q2': "Who was the Roman Emperor when Jesus was born?"

The single multi-hop question is therefore the rewritten 'Q2'.
'Q2' = 'Q' = "Who was the Roman Emperor when Jesus was born?"

You MUST apply this structure when asked to merge 'Q1' and 'Q2'.
Now merge 'Q1' and 'Q2' into a single multi-hop bridge question 'Q'.
'Q2' : {question1}
'Q2' : {question2}

Figure 17: Prompt used to merge Q1 and Q2 in MHQA.

---
**Generating Q1 in MHQA.**

Identify one entity in the following text. Come up with a question so that the answer to this question is the entity chosen earlier. The question must be based on the following text. Write your results as 'Question:' and then the question and 'Entity:' and then the entity.

Text: {document_one}

---

Figure 18: Prompt used to generate $Q_1$. $Q_1$ is generated such that its answer $A1 = E$ where $E$ is the entity retrieved.

---
**Generating Q2 in MHQA.**

Come up with a question based on the following text that contains the word:
{entity}

Text: {document_two}

---

Figure 19: Prompt used to generate $Q_2$. $Q_2$ is generated such that its main topic is $E$ where $E$ is the entity retrieved.

