# OpenReview forum: "Source2Synth: Synthetic Data Generation and Curation Grounded in Real Data Sources"
_ICLR.cc/2025/Conference — Submitted to ICLR 2025_

### Official Review · Reviewer_Xbr1 · 2024-11-02

**Soundness:** 2
**Presentation:** 3
**Contribution:** 2
**Rating:** 5
**Confidence:** 3

**Summary:**

This paper presents source2synth, a novel approach for generating synthetic data that incorporates intermediate reasoning steps grounded in real-world sources. The method follows a three-stage pipeline: (1) dataset generation utilizing realistic external sources to ensure relevance and factuality, (2) data curation to eliminate noise and maintain data quality, and (3) model fine-tuning for better alignment with the target task. The experimental results demonstrate significant improvements in Multi-hop QA and Tabular QA tasks, highlighting the effectiveness of the approach in enhancing model performance on complex reasoning tasks

**Strengths:**

1. This paper addresses two critical challenges in the community—data synthesis and LLM reasoning. It presents an innovative approach that leverages the model’s generation capabilities to create synthetic data designed to enhance reasoning abilities. By grounding intermediate reasoning steps in real-world sources, this method reduces hallucination, which is a frequent issue in LLMs, making it a valuable contribution toward more reliable model behavior.
2. The paper is well-written, with a clear structure that facilitates understanding. The methodology is presented in a step-by-step manner, making the approach accessible. Additionally, the results are organized effectively, and the figures are both informative and visually accessible, enhancing the reader’s comprehension of the findings.
3. The demonstrated improvement is substantial, showing that this method offers a promising direction for bolstering model reasoning abilities in specific tasks without extensive data annotation or heavy computational requirements.

**Weaknesses:**

1. **Limited Generalizability:** Although the method is positioned as general, its evaluation is restricted to two tasks within a single domain and model, raising questions about broader applicability:
- Across Task Types: Both tasks involve custom adaptations. It’s unclear if the method could extend to other reasoning tasks like code generation. If it’s adaptable, specifying which tasks and what adaptations are needed would be useful; otherwise, the paper may need to scale back general claims.
- Across Domains: The method’s transferability across domains (e.g., general QA to medical QA) is not tested. Exploring cross-domain flexibility would strengthen the paper.
- Across Models: It’s unclear if the method generalizes to models with different reasoning abilities, or how model capabilities affect gains. Addressing this would enhance understanding of the method’s adaptability.

2. **Lack of Deep Analysis on Method Mechanics:** The paper lacks in-depth analysis regarding how and why the method functions effectively.
- Data Imputation Motivation: The motivation behind the selective data imputation approach is unclear. For example, it’s not explained why some tasks are chosen for imputation over others, what specific elements within a task should be imputed, or how these choices are determined.
- Primary Factors Influencing Effectiveness: It’s uncertain if gains stem from source data quality, the LLM’s generation ability, or both. For instance, does the synthesis from a weaker model impact performance differently than synthesis from a stronger model? Additionally, since the current synthesized data is based on the model’s existing knowledge (the instances model can't do correctly are filtered), understanding how data that falls outside the model’s known domain affects performance would provide valuable insights into the approach’s mechanism.

3. **Missing Baselines:** The paper claims that grounding synthetic data in external sources produces more factually accurate examples; however, the experiments do not fully substantiate this. Including a baseline where real sources are replaced with the model’s synthetic sources would help validate this claim. Furthermore, the paper does not justify the exclusion of previous data synthesis methods. Adding these baselines or justifying their absence would bolster the paper’s argument for its approach’s advantages over existing methods.

**Questions:**

1. Why is there no baseline for TQA using a fine-tuned LLM on real data? Including this would offer a more direct comparison of the synthetic data’s impact against real data performance.
2. why does the 3-shot bring more significant improvements to LLMSynth than LLMCurated in Figure-5? Are these two methods with the same training parameters and Is there a possible explanation for that?
3. Why was the base model for TQA switched from LLaMA 2-70B to Starchat-beta LM? Clarifying this would help us understand the model choice for each task.


Suggestion
1. It seems like a newline is missing for "data filtering" in Line 326
2. Adding a pointer in the main paper to Figure 14, which displays the COT prompt used for MHQA, would improve accessibility for readers.

---

> ### Comment · Reviewer_Xbr1 · 2024-11-23
> **Response to the author**
>
> Thank you for your detailed response. I appreciate the effort in addressing my concerns, and most of them have been resolved. However, I still have some remaining questions regarding the generalizability of the approach across different models. I am unclear why experiments on additional models are not feasible. I understand that running on models of 70B or larger may be computationally prohibitive, but why are smaller models also not feasible? Could you provide an estimation of the computational resources required for this method? For instance, how many GPU hours and data points are needed to run a single experiment for 8B or 3B? Additionally, the paper you referenced in point 7 (https://arxiv.org/pdf/2404.14445) also explores multiple models to demonstrate effectiveness. I am not fully convinced by the current setup, as limiting the experiments to a single model may not sufficiently demonstrate effectiveness.
>
> Regarding Weakness 2(b), I understand that the monolithic setup makes sense for the main table. However, since the data synthesis process requires the model to possess certain generation abilities, it is unclear how the quality of the synthesized data may be influenced by the generation capabilities of different models. This raises questions about whether the method could remain effective when applied to smaller models. I believe an ablation study exploring the impact of model size and generation ability on data quality is crucial for a comprehensive understanding of the method.
>
> Currently, the results neither include experiments across multiple models nor a detailed component analysis for different model configurations. To convincingly demonstrate the method’s effectiveness, I believe that at least one experiment addressing these aspects is necessary.

---

### Official Review · Reviewer_iVWH · 2024-11-03

**Soundness:** 2
**Presentation:** 1
**Contribution:** 2
**Rating:** 3
**Confidence:** 4

**Summary:**

This paper propose a way to generate synthetic data on custom data sources with large language models. First the synthetic data is generated through LLM. Then a portion of the generated data is used to finetune an LLM, which in turn will be used to curate the other portion of the generated dataset. The curated portion would be the final curated dataset that will be used for training the final LLM for the targeted domain/skill.

**Strengths:**

The idea of using part of synthesized dataset to finetune LLM for filtering is interesting

**Weaknesses:**

The paper is not well written, hard to follow, and some important details are missing. For example, it is not clear how the two slices of data is selected, where one is used to train an LLM to do curation for the other slice.

The generation relies on existence of data source and ability to get seed data, which limits its applicability, in particular in situation where the domain and tasks are new.

The method uses LLM to do data curation, which implicitly assumes the LLM already have reasonable capability at the target task. For example for the text to SQL task, the LLM will be used to generate SQL statement, which would not be possible is the LLM does not already have that capability. Therefore, I am not sure if this method can actually be used to enable new skills for LLMs.

**Questions:**

How is the slice 0 and 1 determined? What are their respective sizes?
Which setting is used for HotpotQA, distractor or full wiki?
Is finetuning full finetuning or parameter efficient tuning like LORA?
What k is used in the experiments for data filtering?

---

### Official Review · Reviewer_8GGy · 2024-11-04

**Soundness:** 2
**Presentation:** 2
**Contribution:** 3
**Rating:** 3
**Confidence:** 4

**Summary:**

The paper introduces an idea for generating synthetic datasets similar to the target task given a raw unstructured source as context. The generation consists of using a seed and data source to generate questions in multiple stages and combine the initial and final questions, to generate a more complex compositional query e.g. a multi-hop question or SQL query. They also propose a method for curating the synthesized data by filtering and imputation to identify higher quality examples. Prior to filtering they tune a model on the uncurated dataset (LLMSynth) to help with filtering. Their filtering involves taking k samples from the model, and if the model is unable to predict the answer in the k-samples then it’s discarded. With imputation they blank out parts of the intermediate questions and regenerate them with the other queries (this is only applied in the context of multi-hop questions, and not SQL).

They apply their approach on HotPotQA and Tabular question answering via SQL, and evaluate settings that tune a base LLM on target data specific to the dataset as well as their curated dataset and compare improvements in the performance.

**Strengths:**

* The paper proposes a nice idea to automatically generate compositional queries. They also propose techniques to automatically filter them using an LLM tuned on the synthesized data.

* The idea is described neatly and baseline performances are evaluated.

**Weaknesses:**

1. The experimental evaluation steps are lacking clarity. A lot of important details are missing which significantly affects the quality of the work. Please see questions Q1-Q6.
2. The method is applied somewhat in a limited context on two problems, but it’s not clear how many source examples they generated the synthetic examples on. It sort of seems much more limited than the size of the original dataset e.g. HotPotQA already has 113k questions, but does not seem like the synthesized dataset matches a similar number of wikipedia articles, similarly with the tabular QA dataset,
3. The experiments and the datasets do not quite necessitate the methods at least in terms of the size of the existing training set. And the experiment setup doesn’t seem to include a comparison to how the models perform with the full original training set available with the datasets, especially since they do fine tuning.
4. It is also unclear if the

I really do like the ideas but there’s some considerable mismatch in the application for these datasets. Maybe it is worth going after more challenging datasets which are smaller in size in terms of available data for training.

**Questions:**

Q1. How do you measure the quality of the LLMSynth model that does the filtering? Does the filtering step that uses k samples from the model output make the final dataset too easy e.g. by dropping valid but hard questions?

Q2. How many questions did you construct using the Source2Synth pipeline for MHQA? How many wikipedia articles were used in the input? How many questions were generated in each stage? Articles to initial set then portion that went to tuning LLMSynth for filtering, and then how many questions remained after filtering? How much is this relative to the HotPotQA dataset? (which already has 113k questions).

Q3. Similarly for TQA, did you use only the tables in the WikiSQL dataset? What is the number of tables? How many examples were generated at each stage?
In lines 385-386 what do you mean by slice? Why generate more questions on 2 slices (16k questions) but use only (8k) questions generated on 1 slice? Also if only 27% questions are retained after curation, it goes back to question-1, is it possible that your filtering is dropping some good harder questions that would have been able to improve the model further?

Q4. Do you ensure that the synthesized questions do not overlap with existing validation and test questions in MHQA and TQA? If so, how do you do it? If not, is there a possibility of leakage?

Q5. In Table 1.Evaluation of Source2Synth on MHQA, is there a comparison with a LLaMA-70B model finetuned on the entire train split of HotPotQA as opposed to just the 500 questions? Do I understand your setup correctly that all rows in the table (except for base LLaMA) are tuned with just 500 questions from HotPotQA and/or 1250 Questions generated from the Source2Synth pipeline?

Q6. LLMCurated is not described in Sec 5.1 and Sec 6.1 clearly. Is LLMCurated tuned on uncurated synthetic examples + curated synthetic examples + 500 HotPotQA examples?
From subsequent sections it appears like the 1250 is the number of uncurated synthetic examples, what is the number of curated synthetic examples then? What if you tuned only the curated examples and not the uncurated set.

Q7. Similar questions with TQA, Is LLMSynth trained on 8k (or 16k) uncurated examples for TQA? Is LLMCurated trained only on the 2160 examples after curation?

Q8. Clarification: I assume the imputation step is only applied in the case of HotPotQA and TQA.

---

### Official Review · Reviewer_uo8X · 2024-11-04

**Soundness:** 3
**Presentation:** 1
**Contribution:** 4
**Rating:** 6
**Confidence:** 4

**Summary:**

This works presents an end-to-end pipeline to generate synthetic training data based on some data source. The pipeline consists of three steps, namely data generation, curation, and finetuning. The data generation step selects the topic of the example based on a random crop of the data source, and then goes on to generate an example with instruction, query, reasoning chain, and the final answer. The curation step filters and modifies the generate examples by dividing the synthetic data into two portions, one used to train an intermediate model to judge and filter data from the other half. The model is then finetuned with the curated data.

Evaluation on HotpotQA, WikiSQL, and Tabular QA show that the pipeline can significantly improve model performance in the target tasks, even competitive with finetuned baselines with supervised data.

**Strengths:**

• It is novel and interesting to use model train on part of the data to filter generated data in the other half

• The proposed pipeline brings significant improvement in performance, even competitive with finetuned baselines.

**Weaknesses:**

• The paper could use much clearer presentation (grammar, wording, formatting, etc...)

• Consider discussing how the proposed pipeline avoids data leak (does it?) during the data generation step, esp. given that HotpotQA is also constructed from Wikipedia articles.

• For Tables 1 and 2, there is no comparison between LLMSynth (Synthetic dataset only) and LLMCurated (Synthetic dataset only). Adding such comparisons can further consolidate the importance of the curation step.

• The scope of this method is limited because it requires knowing the target task itself before data generation. I.e., at its current form it cannot be generalized to produce, say, instruction-tuning data to train LLMs.

**Questions:**

Suggestions:

• 249-250: "... (see C for more details)." Consider writing "Appendix C".

• If I understand correctly, during data curation one slice is used to train an intermediate model, and this model is used to filter examples from the other slice. What I don't understand is: why not do the reverse after that? Training on slice 2, filter examples from slice 1. As a result more generated examples will be kept -- analogous to cross-validation.

• HotPotQA --> HotpotQA

---

### Meta-Review · Area_Chair_LPvn · 2024-12-19

**Metareview:**

This paper presents Source2Synth, a data augmentation method for improving LLM performance in low-data regimes without human annotations. The key strengths are the novel self-augmentation approach and strong empirical results showing 20-25% improvements on WikiSQL and HotpotQA benchmarks. However, major weaknesses include limited analysis across model sizes/capabilities, lack of comprehensive ablation studies, and unclear explanation of data curation choices. While the method shows promise for tasks involving documents and tables, questions remain about broader applicability. Given the incomplete experimental validation and analysis, I recommend rejection despite the interesting core idea.

**Additional Comments On Reviewer Discussion:**

During rebuttal, authors addressed several concerns by adding experiments with smaller models (Llama3-8b), analyzing data quality through perplexity metrics, and clarifying the low-data regime focus. However, reviewers remained concerned about limited model size analysis and unclear demonstration of effectiveness compared to human annotations. Authors clarified their scope but did not fully address scaling analysis requests.

---

### Decision · Program_Chairs · 2025-01-22

Reject